# Fermionic higher-form symmetries

**Yi-Nan Wang[1,2]⋆ and Yi Zhang[2]†**

**1** School of Physics, Peking University, Beijing 100871, China
**2** Center for High Energy Physics, Peking University, Beijing 100871, China

⋆ ynwang@pku.edu.cn , † yi.cheung@pku.edu.cn

## Abstract

In this paper, we explore a new type of global symmetries — the fermionic higher-form symmetries. They are generated by topological operators with fermionic parameter, which act on fermionic extended objects. We present a set of field theory examples with fermionic higher-form symmetries, which are constructed from fermionic tensor fields. They include the free fermionic tensor theories, a new type of fermionic topological quantum field theories, as well as the exotic 6d (4,0) theory. We also discuss the gauging and breaking of such global symmetries and the relation to the no global symmetry swampland conjecture.

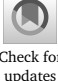

# 1   Introduction

In recent years, a lot of progress has been made in the rapidly evolving field of generalized global symmetries, with fruitful applications in both high energy and condensed matter physics. These include higher-form symmetries which act on extended objects, higher-group symmetries which combine symmetries of different form degrees, and more generally, non-invertible and higher-categorical symmetries [1–3].

In this paper, we explore the possibilities of *fermionic p-form global symmetries* in $d$-dimensional quantum field theories, which act on $p$-dimensional fermionic extended operators. They are generated by $(d-p-1)$-dimensional topological operators with fermionic parameters (spinors with Grassmannian components). We will be restricting ourselves to the cases of invertible fermionic global symmetries.

One simple example is the case of a free Rarita-Schwinger field $\psi_\mu$, as we explain in section 2. Similar to the $U(1)$ Maxwell theory, we can define a gauge invariant "Wilson loop" operator by integrating $\psi_\mu$ along a circle $\mathcal{C}$. The operator is charged under a $(d-2)$-dimensional topological operator. We also extend the discussion to the cases of a free fermionic $p$-form gauge field (first defined in [4–6], and whose properties were recently discussed in [7]) and a more general type of actions with fermionic tensor gauge fields and a single derivative in each term.

Another playground for $p$-form fermionic symmetry is the 6d (4,0) theory, which is conjectured to be the UV completion of strongly coupled 5d maximal supergravity [8]. In particular, we consider the free limit of 6d (4,0) theory with a free kinetic term for the fermionic 2-form gauge field (the exotic gravitino) $\Psi_{MN}$ ($M,N = 0,\ldots,5$) in the 6d (4,0) supermultiplet. We show that $\Psi_{MN}$ possesses a fermionic 2-form symmetry in this limit. We also discuss the dimension reduction of the free action on $S^1$. The fermionic 2-form symmetry is reduced to a fermionic 2-form symmetry and a fermionic 1-form symmetry in 5d, however only one of them is preserved after integrating out a 5d fermionic gauge field.

Analogous to the bosonic higher-form symmetries, we consider the gauging of fermionic $p$-form global symmetries by coupling to a background fermionic $(p+1)$-form gauge field, in section 4. Nonetheless, unlike the case of bosonic $p$-form symmetry, we found that even for the case of a free fermionic $p$-form gauge field, an 't Hooft anomaly exists which obstructs the gauging of fermionic $p$-form global symmetries. On the other hand, we also present a few cases where the fermionic symmetries can be gauged.

One may also wonder what is the fate of the fermionic $p$-form global symmetry after the corresponding fermionic $p$-form gauge field is coupled to other fields. We discuss the case of a charged Rarita-Schwinger field under a bosonic $U(1)$ gauge group as well as the case of supergravity in section 5. In these cases, the fermionic 1-form global symmetry is explicitly broken by the coupling terms with the $U(1)$ gauge fields $A_\mu$ or the Christoffel connection. We also briefly discuss the relation to the no global symmetry swampland conjecture [9–11].

The structure of the paper is as follows: in section 2 we introduce the notion of fermionic $p$-form symmetries and discuss the examples of free fermionic $p$-form gauge fields. In section 3 we discuss examples with unbroken fermionic $p$-form symmetries, including a generalized fermionic "BF" theory and the free limit of 6d (4,0) theory. In section 4 we discuss the gauging of fermionic $p$-form symmetries. In section 5 we present a few examples where the fermionic symmetries are explicitly broken, and discussed the swampland implications. Finally, we put some discussions and future directions in section 6. The conventions in the formula are organized in appendix A.

## 2 Fermionic higher-form symmetries

We introduce the notion of fermionic $p$-form symmetries, which are the fermionic counterparts of the $p$-form symmetries acting on bosonic $p$-dimensional operators [1]. We work in $d$-dimensional Minkowski space-time[1] $\mathbb{R}^{1,d-1}$, and the results can be applied to Euclidean cases as well.

An *invertible fermionic p-form symmetry* is generated by topological operators $U_\epsilon(M^{(d-p-1)})$, defined on a $(d-p-1)$-dimensional submanifold $M^{(d-p-1)} \subset \mathbb{R}^{1,d-1}$. The symmetry parameter $\epsilon \in G$ is a fermionic spinor, i.e. it has anti-commuting (Grassmannian) components. The symmetry group $G$ is also fermionic, which can be interpreted as the non-compact translation group of spinors with Grassmannian components, namely $\mathbb{R}^s$, where $s$ is the number of spinor components. Thus, it is always Abelian. $U_\epsilon(M^{(d-p-1)})$ acts on $p$-dimensional operators $V(\mathcal{C}^{(p)})$ as

$$\langle U_\epsilon(M^{(d-p-1)})V(\mathcal{C}^{(p)})\rangle = R(\epsilon)^{\langle \mathcal{C}^{(p)}, M^{(d-p-1)}\rangle}\langle V(\mathcal{C}^{(p)})\rangle. \tag{1}$$

$\langle \mathcal{C}^{(p)}, M^{(d-p-1)}\rangle$ is the linking number (cf. (A.13)) between $\mathcal{C}^{(p)}$ and $M^{(d-p-1)}$. $R$ is a representation of the symmetry group $G$.

As usual, when the symmetry parameter $\epsilon$ is space-time independent, we call it a fermionic $p$-form global symmetry. Otherwise it is a gauge symmetry.

Now we discuss some examples.

**Fermionic 0-form symmetry.** A common example of fermionic 0-form symmetry is the *global supersymmetry*, with a spinor $\epsilon$ as the symmetry parameter. The $(d-1)$-dimensional topological operator is

$$U_\epsilon(M^{(d-1)}) = \exp\left(\int_{M^{(d-1)}} i(\bar{\epsilon}\mathcal{J} + \bar{\mathcal{J}}\epsilon)\right)$$
$$= e^{i(\bar{\epsilon}Q + \bar{Q}\epsilon)}. \tag{2}$$

$\mathcal{J}$ is the supercurrent $(d-1)$-form and $Q$ is the supercharge, which acts on the space of local operators.

As another simple example, let us consider a free Dirac spinor $\psi$ with the action

$$S = \int -\bar{\psi}\gamma^\mu \partial_\mu \psi d^d x. \tag{3}$$

This action admits a *fermionic shift symmetry*

$$\psi \to \psi + \epsilon, \tag{4}$$

where the spinor parameter $\epsilon$ satisfies $d\epsilon = 0$. It is a fermionic 0-form symmetry generated by the topological operator

$$U_\epsilon(M^{(d-1)}) = \exp\left(i\int_{M^{(d-1)}} \frac{1}{(d-1)!}\varepsilon_{\mu_1\cdots\mu_{d-1}\nu}\left[\bar{\epsilon}\gamma^\nu\psi - \bar{\psi}\gamma^\nu\epsilon\right]dV^{\mu_1\cdots\mu_{d-1}}\right), \tag{5}$$

where $\varepsilon$ is the Levi-Civita symbol in $d$ dimensions and $dV^{\mu_1\cdots\mu_{d-1}}$ is the volume element on the $(d-1)$-dimensional submanifold $M^{(d-1)}$.
Conveniently, $U_\epsilon(M^{(d-1)})$ can be written compactly using differential form notation

$$U_\epsilon(M^{(d-1)}) = \exp\left(i\int_{M^{(d-1)}} \star\left[\bar{\epsilon}\gamma_{(1)}\psi - \bar{\psi}\gamma_{(1)}\epsilon\right]\right), \tag{6}$$

---

[1]The fermionic $p$-form symmetry exists on any flat space-time as well, such as the torus. We briefly comment on the cases of curved space-time manifold in section 5.2, but we leave the detailed analysis in future work.

where $\star$ is the Hodge star operator. The combination $\bar\epsilon \gamma_{(1)}\psi$ can be regarded as a 1-form $\bar\epsilon \gamma_{(1)}\psi = \bar\epsilon \gamma_\mu \psi \, dx^\mu$. The Dirac equation $\slashed\partial \psi = 0$ is translated into $d \star \gamma_{(1)}\psi = 0$, and together with Stokes' theorem we see that $U_\epsilon(M^{(d-1)})$ is on-shell topological.

As another comment, note that we can relax the spinor parameter $\epsilon$ in (4) to only satisfy a weaker constraint, i. e. the Dirac equation $\slashed\partial \epsilon = 0$. Now the parameter $\epsilon$ is space-time dependent in general, but the action (3) is still invariant. In this case, we say the shift symmetry (4) is a "semi-local" symmetry.[2]

**Fermionic 1-form symmetry.** The first new example is the case of fermionic 1-form symmetry. Let us consider a free Rarita-Schwinger field $\psi_\mu$ in $d \geq 3$ with the action

$$S[\psi_\mu] = \int -\bar\psi_\mu \gamma^{\mu\nu\rho} \partial_\nu \psi_\rho \, d^d x \,. \tag{7}$$

For $d \geq 4$, $\psi_\mu$ is fermionic due to spin-statistics theorem, while there is no restriction in $d = 3$. The equation of motion for $\psi_\mu$ is

$$\gamma^{\mu\nu\rho} \partial_\nu \psi_\rho = 0 \,, \tag{8}$$

which is equivalent to[3]

$$\gamma^\mu H_{\mu\nu} = 0 \,, \tag{9}$$

where we define the field strength $H_{\mu\nu}$ as exterior derivative of the Rarita-Schwinger field: $H_{\mu\nu} = 2\partial_{[\mu}\psi_{\nu]}$. The action has a fermionic 0-form gauge symmetry

$$\psi_\mu \to \psi_\mu + \partial_\mu \lambda \,, \tag{10}$$

and the gauge transformation parameter $\lambda$ is a spinor. The field strength $H_{\mu\nu}$ is gauge invariant.

Now we construct the gauge invariant 1-dimensional topological operator[4]

$$V_\eta(\mathcal{C}) = \exp\left( i \oint_{\mathcal{C}} (\bar\eta \psi_\mu + \bar\psi_\mu \eta) \, dx^\mu \right) = \exp\left( i \int_{\mathcal{C}} (\bar\eta \psi_{(1)} + \bar\psi_{(1)} \eta) \right), \tag{11}$$

as the fermionic analog of Wilson loop for the 1-form gauge field $A_\mu$, and $\psi_{(1)} = \psi_\mu \, dx^\mu$ is the fermionic one-form. The spinor parameter $\eta$ is an analogue of the charge for a bosonic Wilson loop. In our case, since $\psi_\mu$ is a non-compact gauge field, $\eta$ do not need to be quantized.

We define a fermionic 2-form $\mathcal{J}_{(2)} = \frac{1}{2}\mathcal{J}_{\mu\nu}\, dx^\mu \wedge dx^\nu$ whose components are given as

$$\mathcal{J}_{\mu\nu} \equiv \gamma_{\mu\nu\rho}\psi^\rho \,. \tag{12}$$

By the equation of motion (8), this two form is conserved

$$\partial^\mu \mathcal{J}_{\mu\nu} = -\gamma_{\nu\mu\rho} \partial^\mu \psi^\rho = 0 \,. \tag{13}$$

This fact indicates that we can define a closed $(d-2)$-form current $(\star\mathcal{J})_{(d-2)}$ (analogous to the free Maxwell theory, cf. $\partial^\mu F_{\mu\nu} = 0$) and the closure of this $(d-2)$-form current ensures the topological feature of the $(d-2)$-dimensional operator $U_\epsilon(M^{(d-2)})$

$$U_\epsilon(M^{(d-2)}) = \exp\left( i \int_{M^{(d-2)}} \left[ \bar\epsilon (\star\mathcal{J})_{(d-2)} + (\star\bar{\mathcal{J}})_{(d-2)} \epsilon \right] \right), \tag{14}$$

---

[2]In an analogous setup, a free non-compact scalar with action $S = \int_{M_d} \partial_\mu \phi \partial^\mu \phi$ also has a shift symmetry $\phi \to \phi + a$, where the action is invariant if $a$ only satisfies the weaker condition $\partial_\mu \partial^\mu a = 0$. This is known as the "Galileon symmetry" [12].

[3]In $d = 3$, this reduces to $H_{\mu\nu} = 0$, which means that $\psi_\mu$ has no physical degrees of freedom. We will discuss these topological fermionic $p$-forms in a more general context in section 3.

[4]We also analyse the behaviour of the vacuum expectation value of the fermionic Wilson loop in the appendix B.

where $\bar{\mathcal{J}}_{\mu\nu} = \bar{\psi}^\rho \gamma_{\mu\nu\rho}$. In components, it reads

$$U_\epsilon(M^{(d-2)}) = \exp\left(i\left[\bar{\epsilon}\int_{M^{(d-2)}}\frac{1}{(d-2)!}\frac{1}{2!}\varepsilon_{\mu_1\dots\mu_{d-2}\mu\nu}\gamma^{\mu\nu\rho}\psi_\rho\, dV^{\mu_1\dots\mu_{d-2}} + \text{c.c.}\right]\right). \quad (15)$$

Let us check the action of $U_\epsilon(M^{(d-2)})$ on $V_\eta(\mathcal{C})$ and verify

$$\langle U_\epsilon(M^{(d-2)})V_\eta(\mathcal{C})\rangle = \exp\left(i(\bar{\epsilon}\eta + \bar{\eta}\epsilon)\langle\mathcal{C}, M^{(d-2)}\rangle\right)\langle V_\eta(\mathcal{C})\rangle, \quad (16)$$

where $\langle\mathcal{O}\rangle$ means the vacuum expectation value of an operator $\mathcal{O}$. The left-hand side of (16) is by definition computed as the path integral

$$\langle U_\epsilon(M^{(d-2)})V_\eta(\mathcal{C})\rangle = \int \mathcal{D}\psi_\mu\, \mathcal{D}\bar{\psi}_\mu\, e^{iS[\psi_\mu,\bar{\psi}_\mu] + i\int_{M^{(d-2)}}\left(\bar{\epsilon}(\star\mathcal{J})_{(d-2)} + (\star\bar{\mathcal{J}})_{(d-2)}\epsilon\right) + i\int_{\mathcal{C}}(\bar{\eta}\psi_{(1)} + \bar{\psi}_{(1)}\eta)}. \quad (17)$$

As a starting point, we consider one of the $(d-2)$-form integrals

$$\int_{M^{(d-2)}}\bar{\epsilon}(\star\mathcal{J})_{(d-2)} = \int_{M^{(d-1)}}\bar{\epsilon}(d\star\mathcal{J})_{(d-1)} = \int_{M^{(d)}}J_{(1)}(M^{(d-1)})\wedge\bar{\epsilon}(d\star\mathcal{J})_{(d-1)}$$
$$= \int\bar{\epsilon}J_\mu\gamma^{\mu\nu\rho}\partial_\nu\psi_\rho\, d^d x. \quad (18)$$

Here in the first step we used Stokes' theorem and $\partial M^{(d-1)} = M^{(d-2)}$. In the second step, we introduced a one-form delta-function current $J_{(1)}$ with support on the $M^{(d-1)}$ submanifold. Explicitly, it is given in local coordinates as[5]

$$J_\mu(x, M^{(d-1)}) = \int_{M^{(d-1)}}\frac{1}{(d-1)!}\varepsilon_{\mu\mu_1\dots\mu_{d-1}}\delta^{(d)}(x-y)\, dV^{\mu_1\dots\mu_{d-1}}(y). \quad (19)$$

This is quite similar to the delta-function differential forms (cf. (A.12)) used in describing $D$-brane (Membrane) sources in String (M-) theory and mathematically it is the cohomology class of the Poincaré dual of $M^{(d-1)}$ [14]. We use this trick to write the $(d-2)$-form integral as an integral on the entire manifold $M^{(d)}$ and this will help us to absorb it into the free action by a change of variables.

We now take a look back at the path integral and under the change of variable $\psi_\mu \to \psi'_\mu = \psi_\mu - \epsilon J_\mu$ the action becomes

$$S[\psi_\mu - \epsilon J_\mu] = \int -\left(\bar{\psi}_\mu - \bar{\epsilon}J_\mu\right)\gamma^{\mu\nu\rho}\partial_\nu\left(\psi_\rho - \epsilon J_\rho\right)d^d x$$
$$= S[\psi_\mu] + \int\left(\bar{\psi}_\mu\gamma^{\mu\nu\rho}\epsilon\partial_\nu J_\rho + \bar{\epsilon}J_\mu\gamma^{\mu\nu\rho}\partial_\nu\psi_\rho - J_\mu\bar{\epsilon}\gamma^{\mu\nu\rho}\epsilon\partial_\nu J_\rho\right)d^d x.$$

The last term can be regularized by a local counter term [15]. The third term is identified with the RHS of (18), and similarly the second term is identified with the hermitian conjugate of the third term. (cf. (14))

Apply this change of variable in the path integral (17), we get

$$\langle U_\epsilon(M^{(d-2)})V_\eta(\mathcal{C})\rangle = e^{i(\bar{\epsilon}\eta + \bar{\eta}\epsilon)\oint_{\mathcal{C}}J_\mu dx^\mu}\int\mathcal{D}\psi'_\mu\,\mathcal{D}\bar{\psi}'_\mu\, e^{iS[\psi'_\mu,\bar{\psi}'_\mu] + i\int_{\mathcal{C}}(\bar{\eta}\psi'_{(1)} + \bar{\psi}'_{(1)}\eta)}$$
$$= \exp\left(i(\bar{\epsilon}\eta + \bar{\eta}\epsilon)\langle\mathcal{C}, M^{(d-2)}\rangle\right)\langle V_\eta(\mathcal{C})\rangle. \quad (20)$$

---

[5]We are using a slightly different definition from [13], since for us the Poincaré dual $J_{(d-p)}(\mathcal{C}^{(p)})$ of a $p$-cycle $\mathcal{C}^{(p)}$ is characterized by $\int_{\mathcal{C}^{(p)}}A_{(p)} = \int_{M^{(d)}}J_{(d-p)}(\mathcal{C}^{(p)})\wedge A_{(p)}$ for any $p$-form $A_{(p)}$ instead of by $\int_{\mathcal{C}}A = \int_M A\wedge J$.

Recall that $\partial M^{(d-1)} = M^{(d-2)}$, and $\langle \mathcal{C}, M^{(d-2)} \rangle$ is the linking number of $\mathcal{C}$ and $M^{(d-2)}$

$$\langle \mathcal{C}, M^{(d-2)} \rangle = \mathcal{I}(\mathcal{C}, M^{(d-1)}) = \oint_{\mathcal{C}} J_{(1)}(M^{(d-1)}). \tag{21}$$

The action of the fermionic 1-form symmetry on the original Rarita-Schwinger field $\psi_\mu$ is the shift by a fermionic 1-form $\xi_\mu$[6]

$$\begin{aligned} \psi_\mu &\longrightarrow \psi_\mu + \xi_\mu, \\ \partial_{[\nu}\xi_{\rho]} &= 0. \end{aligned} \tag{22}$$

One can also define a topological operator generating a $(d-3)$-form magnetic symmetry, similar to the Maxwell theory, using the field strength $H$ of the Rarita-Schwinger field

$$U_\theta(M^{(2)}) = \exp\left( i \int_{M^{(2)}} \left[ (\bar{\theta} H)_{(2)} + \text{c.c.} \right] \right), \tag{23}$$

and $H$ satisfies the Bianchi identity $dH = 0$. However, the charged object under such symmetry is not clear. In the bosonic case, one finds 't-Hooft lines via electromagnetic duality as charged operators under the magnetic higher-form symmetry. We have not yet found EM-type dualities between fermionic $p$-forms,[7] and we will investigate it in a future work.

**Fermionic $p$-form symmetry.** The flat spacetime Rarita-Schwinger action (7) for a fermionic one-form $\psi_\mu$ can be generalised to a free action for 'fermionic $p$-forms' $\psi_{(p)}$ [4–7]

$$S[\psi_{(p)}] = -(-1)^{\frac{p(p-1)}{2}} \int d^d x \, \bar{\psi}_{\mu_1\mu_2...\mu_p} \gamma^{\mu_1\mu_2...\mu_p \nu \rho_1\rho_2...\rho_p} \, \partial_\nu \psi_{\rho_1\rho_2...\rho_p}. \tag{24}$$

This action is invariant under the gauge transformation

$$\delta\psi_{\mu_1\mu_2...\mu_p} = p \, \partial_{[\mu_1}\Lambda_{\mu_2...\mu_p]}, \tag{25}$$

and $\Lambda_{\mu_2...\mu_p}$ is the components of the fermionic $(p-1)$-form $\Lambda_{(p-1)}$ gauge parameter. The gauge transformation can be written compactly as $\delta\psi_{(p)} = d\Lambda_{(p-1)}$. We define the gauge invariant field strength $H_{(p+1)}$ of the field $\psi_{(p)}$ as $H_{(p+1)} \equiv d\psi_{(p)}$, i. e.

$$H_{\mu_1...\mu_{p+1}} = (p+1) \partial_{[\mu_1}\psi_{\mu_2...\mu_{p+1}]}. \tag{26}$$

The equations of motion derived from the action (24) read

$$\gamma^{\mu_1...\mu_p \nu_1...\nu_{p+1}} H_{\nu_1...\nu_{p+1}} = 0, \tag{27}$$

which after some gamma matrices identities are equivalent to the single-gamma-trace equations

$$\gamma^{\mu_1} H_{\mu_1\mu_2...\mu_{p+1}} = 0. \tag{28}$$

The latter cannot be obtained directly from an action. For $2p+1 > d$, the action (24) identically vanishes because of the rank of the gamma matrix exceeds the highest possible value $d$, while for $2p+1 = d$, the equations of motion are equivalent to $H_{(p+1)} = 0$, which implies that $\psi_{(p)}$ is a pure gauge. The only case the theory described by (24) has propagating degrees of freedom is when $2p < d$.

Analogous to the 1-form case, we define a conserved $(p+1)$-form $\mathcal{J}_{(p+1)}$ with components

$$\mathcal{J}_{\mu_1\mu_2...\mu_{p+1}} \equiv \gamma_{\mu_1...\mu_{p+1}\nu_1...\nu_p} \psi^{\nu_1...\nu_p}, \tag{29}$$

and $\partial_\mu \mathcal{J}^{\mu\mu_1...\mu_p} = 0$ is achieved by requiring equation of motions (27).

---

[6]Just as we discussed in the 0-form case, one can relax the flatness condition of $\xi_\mu$, i.e. require it to be "flat" under the Rarita-Schwinger differential operator $\gamma^{\mu\nu\rho}\partial_\nu$ instead of the usual space-time derivative.

[7]See [16] for an early attempt on this subject.

We can then build topological operators on $(d-p-1)$-dimensional submanifolds $M^{(d-p-1)}$

$$U_\epsilon(M^{(d-p-1)}) = \exp\left(i\,C(d,p)\int_{M^{(d-p-1)}}\left[\bar\epsilon\,(\star\mathcal{J})_{(d-p-1)} + (\star\bar{\mathcal{J}})_{(d-p-1)}\epsilon\right]\right), \qquad (30)$$

which act on the following $p$-dimensional objects

$$V_\eta(\mathcal{C}^{(p)}) = \exp\left(i\int_{\mathcal{C}^{(p)}}(\bar\eta\psi_{(p)} + \bar\psi_{(p)}\eta)\right), \qquad (31)$$

and the numerical constant is computed (A.14) as $C(d,p) = -p!(-1)^{\frac{(p+1)(2d-p)}{2}}$, it depends on space-time dimension and the degree of the fermionic differential form. The symmetry transformation of $V_\eta(\mathcal{C}^{(p)})$ is a straight generalisation of (16)

$$\langle U_\epsilon(M^{(d-p-1)})V_\eta(\mathcal{C}^{(p)})\rangle = \exp\left(i(\bar\epsilon\eta + \bar\eta\epsilon)\langle\mathcal{C}^{(p)}, M^{(d-p-1)}\rangle\right)\langle V_\eta(\mathcal{C}^{(p)})\rangle. \qquad (32)$$

## 3 Examples with fermionic higher-form global symmetries

### 3.1 Fermionic "BF" theory

In this section, we consider a generalization of the free $p$-form fermionic fields, which consists of an arbitrary number of fermionic $p$-form fields $\psi_{(p),\alpha}$ with different $p$ and label $\alpha$. For the action, it is different for the cases of even and odd space-time dimension $d$.

For odd $d$, it can be written as a sum

$$S = \sum c_i \int_{M^{(d)}} \bar\psi_{(p_i),\alpha_i}\wedge\gamma_{(d-p_i-q_i-1)}\wedge d\psi_{(q_i),\beta_i} + \text{c.c.} \qquad (33)$$

$\gamma_{(d-p_i-q_i-1)}$ is the $(d-p_i-q_i-1)$-form $\gamma$-matrix, i.e. $\gamma_{(r)} = \frac{1}{r!}\gamma_{\mu_1...\mu_r}dx^{\mu_1}...dx^{\mu_r}$. Hence $S$ is the integration of a general sum of $d$-forms, each of which consists of two fermionic fields and a single exterior derivative.

To see how this generalises the free fermionic $p$-form action (24), we consider a single summand

$$\begin{aligned}
S &= c\int_{M^{(d)}} \bar\psi_{(p),\alpha}\wedge\gamma_{(d-p-q-1)}\wedge d\psi_{(q),\beta} + \text{c.c.} \\
&\propto \int d^d x\,\bar\psi_{\mu_1...\mu_p,\alpha}\gamma_{\nu_1...\nu_{d-p-q-1}}\varepsilon^{\mu_1...\mu_p\nu_1...\nu_{d-p-q-1}\rho\sigma_1...\sigma_q}\partial_\rho\psi_{\sigma_1...\sigma_q,\beta} + \text{c.c.} \qquad (34)\\
&\propto \int d^d x\,\bar\psi_{\mu_1...\mu_p,\alpha}\gamma^{\mu_1...\mu_p\rho\sigma_1...\sigma_q}\partial_\rho\psi_{\sigma_1...\sigma_q,\beta} + \text{c.c.},
\end{aligned}$$

where in the last step we used the identity of gamma matrices for odd dimensions

$$\gamma^{\mu_1...\mu_p} = i^{\frac{d+1}{2}}\frac{1}{(d-p)!}\varepsilon^{\mu_1...\mu_p\nu_1...\nu_{d-p}}\gamma_{\nu_{d-p}...\nu_1}. \qquad (35)$$

For even $d$, this identity becomes

$$\gamma^{\mu_1...\mu_p}\gamma_{d+1} = -(-i)^{\frac{d}{2}+1}\frac{1}{(d-p)!}\varepsilon^{\mu_p...\mu_1\nu_1...\nu_{d-p}}\gamma_{\nu_1...\nu_{d-p}}, \qquad (36)$$

where the chirality matrix $\gamma_{d+1}$ (see (A.2) for definition) appears.

The general action takes the form of

$$S = \sum_i c_i \int_{M^{(d)}} \bar{\psi}_{(p_i),\alpha_i} \wedge \gamma_{(d-p_i-q_i-1)}(1 + d_i\gamma_{d+1}) \wedge d\psi_{(q_i),\beta_i} + \text{c.c.} \tag{37}$$

The actions (33), (37) are gauge invariant under the gauge transformations

$$\delta\psi_{(p),\alpha} = d\lambda_{(p-1),\alpha} \,. \tag{38}$$

We prove that there is a fermionic $p$-form global symmetry associated to each fermionic $p$-form field $\psi_{(p),\alpha}$, acting on the $p$-dimensional object

$$V_{\alpha,\eta}(\mathcal{C}^{(p)}) = \exp\left(i\int_{\mathcal{C}^{(p)}} (\bar{\eta}\psi_{(p),\alpha} + \bar{\psi}_{(p),\alpha}\eta)\right) \,. \tag{39}$$

We only need to construct the topological operator generating the fermionic $p$-form symmetry, which takes the form of

$$U_{\alpha,\epsilon}(M^{(d-p-1)}) = \exp\left(i\int_{M^{(d-p-1)}} \left[\bar{\epsilon}\mathcal{K}_{(d-p-1),\alpha} + \bar{\mathcal{K}}_{(d-p-1),\alpha}\epsilon\right]\right) \,. \tag{40}$$

The $(d-p-1)$-form[8] $\mathcal{K}_{(d-p-1),\alpha}$ appears in the equation of motion of $\psi_{(p),\alpha}$ as

$$d\mathcal{K}_{(d-p-1),\alpha} = 0 \,. \tag{41}$$

It is then obvious that $U_{\alpha,\epsilon}(M^{(d-p-1)})$ acts on $V_{\alpha,\eta}(\mathcal{C}^{(p)})$ in the standard way

$$\langle U_{\alpha,\epsilon}(M^{(d-p-1)})V_{\alpha,\eta}(\mathcal{C}^{(p)})\rangle = \exp\left(i(\bar{\epsilon}\eta + \bar{\eta}\epsilon)\langle\mathcal{C}^{(p)}, M^{(d-p-1)}\rangle\right)\langle V_{\alpha,\eta}(\mathcal{C}^{(p)})\rangle \,. \tag{42}$$

Using the general form of the action (33), we can expand out for odd $d$

$$\mathcal{K}_{(d-p-1,\alpha)} = -\sum_{i|\alpha_i=\alpha} c_i\gamma_{(d-p_i-q_i-1)} \wedge \psi_{(q_i),\beta_i} - \sum_{j|\beta_j=\alpha} c_j\gamma_{(d-p_j-q_j-1)} \wedge \psi_{(q_j),\beta_j} \,, \tag{43}$$

and for even $d$

$$\begin{aligned}
\mathcal{K}_{(d-p-1,\alpha)} = &-\sum_{i|\alpha_i=\alpha} c_i\gamma_{(d-p_i-q_i-1)}(1 + d_i\gamma_{d+1}) \wedge \psi_{(q_i),\beta_i} \\
&-\sum_{j|\beta_j=\alpha} c_j\gamma_{(d-p_j-q_j-1)}(1 + d_j\gamma_{d+1}) \wedge \psi_{(q_j),\beta_j} \,.
\end{aligned} \tag{44}$$

When $d = p_i + q_i + 1$ for all $i$, the action (33) describes a non-trivial TQFT (defined on flat manifold) with a rich spectrum of gauge invariant extended operators, which have non-trivial correlation functions (42). Note that comparing with the bosonic TQFT constructed with $p$-form gauge fields, we have many more possibilities from the insertion of $\gamma$-matrices in the action (33).

As a more concrete example, we look at the fermionic analog of the bosonic BF theory:

$$S = \int d^d x \left(-\bar{\chi}_{\mu\nu}\gamma^{\mu\nu\rho\sigma}\partial_\rho\psi_\sigma - \bar{\psi}_\mu\gamma^{\mu\nu\rho\sigma}\partial_\nu\chi_{\rho\sigma}\right) \,. \tag{45}$$

The action is gauge invariant under the 0-form symmetry

$$\delta_\epsilon\psi_\mu = \partial_\mu\epsilon \,, \tag{46}$$

---

[8]In the previous sections, we used its dual $(p+1)$-form $\mathcal{J}_{(d+1)}$.

and the 1-form symmetry

$$\delta_\lambda \chi_{\mu\nu} = 2\partial_{[\mu}\lambda_{\nu]}. \tag{47}$$

We can write down the following gauge invariant observables

$$V_\eta(\mathcal{C}) = \exp\left( i \oint_{\mathcal{C}} (\bar{\eta}\psi_\mu + \bar{\psi}_\mu \eta)\, dx^\mu \right), \tag{48}$$

supported on a loop $\mathcal{C}$ and

$$W_\xi(\mathcal{S}) = \exp\left( i \oint_{\mathcal{S}} \frac{1}{2}(\bar{\xi}\chi_{\mu\nu} + \bar{\chi}_{\mu\nu}\xi)\, dS^{\mu\nu} \right), \tag{49}$$

supported on a surface $\mathcal{S}$.

In $d = 4$ dimensions, both of them are topological[9] and they have non-trivial actions on each other

$$\langle V_\eta(\mathcal{C})W_\xi(\mathcal{S})\rangle = \exp\left( \frac{1}{2}(\bar{\xi}\gamma_5\eta + \bar{\eta}\gamma_5\xi)\langle \mathcal{S},\mathcal{C}\rangle \right) \langle W_\xi(\mathcal{S})\rangle,$$

$$\langle W_\xi(\mathcal{S})V_\eta(\mathcal{C})\rangle = \exp\left( \frac{1}{2}(\bar{\xi}\gamma_5\eta + \bar{\eta}\gamma_5\xi)\langle \mathcal{C},\mathcal{S}\rangle \right) \langle V_\eta(\mathcal{C})\rangle \tag{50}$$

$$= \exp\left( \frac{1}{2}(\bar{\xi}\gamma_5\eta + \bar{\eta}\gamma_5\xi)\langle \mathcal{S},\mathcal{C}\rangle \right) \langle V_\eta(\mathcal{C})\rangle.$$

Note that here the chirality matrix $\gamma_5$ appears, since in 4d the action (45) that we started with is equivalent to

$$S = \int d^d x \left( i\bar{\chi}_{\mu\nu}\varepsilon^{\mu\nu\rho\sigma}\gamma_5\partial_\rho\psi_\sigma + \text{c.c.} \right), \tag{51}$$

due to the identity $\gamma_{\mu\nu\rho\sigma} = -i\varepsilon_{\mu\nu\rho\sigma}\gamma_5$.

For $d > 4$, although $V_\eta(\mathcal{C})$ and $W_\xi(\mathcal{S})$ have a trivial correlation function between each other, we can still construct topological operators acting on them following the general prescriptions above. For even dimensions $d = 2m$

$$U_\epsilon(M^{(d-2)}) = \exp\left( 2(-i)^m \int_{M^{(d-2)}} (\bar{\epsilon}\gamma_{(d-4)}\gamma_{d+1} \wedge \chi_{(2)} + \text{c.c.}) \right),$$

$$U_\lambda(M^{(d-3)}) = \exp\left( 2(-i)^m \int_{M^{(d-3)}} (\bar{\lambda}\gamma_{(d-4)}\gamma_{d+1} \wedge \psi_{(1)} + \text{c.c.}) \right), \tag{52}$$

and for odd dimensions $d = 2m + 1$

$$U_\epsilon(M^{(d-2)}) = \exp\left( 2(-1)^{(m-2)(2m-3)}i^m \int_{M^{(d-2)}} (\bar{\epsilon}\gamma_{(d-4)} \wedge \chi_{(2)} + \text{c.c.}) \right),$$

$$U_\lambda(M^{(d-3)}) = \exp\left( 2(-1)^{(m-2)(2m-3)}i^m \int_{M^{(d-3)}} (\bar{\lambda}\gamma_{(d-4)} \wedge \psi_{(1)} + \text{c.c.}) \right), \tag{53}$$

such that their actions on $V_\eta(\mathcal{C})$ and $W_\xi(\mathcal{S})$ have the same simple form

$$\langle U_\epsilon(M^{(d-2)})V_\eta(\mathcal{C})\rangle = \exp\left( i(\bar{\epsilon}\eta + \bar{\eta}\epsilon)\langle \mathcal{C}, M^{(d-2)}\rangle \right) \langle V_\eta(\mathcal{C})\rangle,$$

$$\langle U_\lambda(M^{(d-3)})W_\xi(\mathcal{S})\rangle = \exp\left( i(\bar{\lambda}\xi + \bar{\xi}\lambda)\langle \mathcal{S}, M^{(d-3)}\rangle \right) \langle W_\xi(\mathcal{S})\rangle. \tag{54}$$

We also briefly discuss the example of a Rarita-Schwinger field $\psi_\mu$ in 3d. In this case the action is

$$S = \int -\bar{\psi}_\mu \gamma^{\mu\nu\rho}\partial_\nu\psi_\rho\, d^3 x. \tag{55}$$

---

[9]In fact, we can also insert $\gamma_5$ in the spinor bilinears in the integrand, i.e. $\bar{\eta}\psi_\mu \to \bar{\eta}\gamma_5\psi_\mu$, to define topological operators.

Since $\gamma^{\mu\nu\rho} = -\epsilon^{\mu\nu\rho}$, the action is topological. In this case there is a fermionic 1-form symmetry associated to $\psi_\mu$. One may also put the theory on a manifold $M^{(3)}$ with boundary $M^{(2)} = \partial M^{(3)}$, for example $M^{(3)} = \mathbb{R}^{1,1} \times \mathbb{R}_{\geq 0}$, $M^{(2)} = \mathbb{R}^{1,1}$. In this case one can repeat the discussion of chiral edge modes (see e. g. [17]) in the case of bosonic Chern-Simons theory. There will be fermionic chiral edge modes propagating on $M^{(2)}$, whose velocity $v$ depends on the boundary condition of $\psi_\mu$ on $M^{(2)}$. Note that these fermionic edge modes are non-compact.

We comment on the spectrum of these TQFTs. In general when we have a $d$-dimensional fermionic TQFT whose Lagrangian is a sum of $d$-forms, the equation of motion for each fermionic $p$-form field $\psi_{(p)}$ is given by $d\psi_{(p)} = 0$. $\psi_{(p)}$ can be interpreted as a pure gauge [7], and it is considered as infinitely massive. Hence the TQFT is a gapped theory.

We can also construct fermionic higher-group-like gauge theories. We take the following 2-group example, where the gauge transformation of $\psi$ (1-form spinor) and $\chi$ (2-form spinor) fields are

$$
\begin{aligned}
\delta\psi &= d\epsilon + \kappa\lambda \,, \\
\delta\chi &= d\lambda \,.
\end{aligned}
\tag{56}
$$

We can construct the following gauge invariant field strengths:

$$
\begin{aligned}
\mathcal{F} &= d\psi - \kappa\chi \,, \\
\mathcal{H} &= d\chi \,,
\end{aligned}
\tag{57}
$$

which are analogues of the fake curvature and (fake) 2-curvatures of the bosonic 2-group gauge theory [18–21].

One can construct gauge invariant actions in $d$ space-time dimensions using $\mathcal{F}$, $\mathcal{H}$ and gamma matrices. These possibilities would be further explored in future works.

## 3.2 6d exotic theories

Fermionic 2-forms appear in exotic six-dimensional supermultiplets [22,23]. These are massless representations of extended Poincaré supersymmetry in parallel to the standard supergravity multiplets. Intriguingly, the highest spin field in these multiplets is either a spin-2 boson which is not a graviton[10] or a "two indexed" exotic gravitino $\Psi_{MN}$ ($M, N = 0, \ldots, 5$) [7, 8, 25]. The $\mathcal{N} = (4,0)$ and $\mathcal{N} = (3,1)$ maximally supersymmetric cases are conjectured by Hull [8, 26–28] to play a role in some strongly coupled regimes of $5d$ maximal supergravity and further studies have been carried out recently [7, 25, 29–38]. Nonetheless similar chiral multiplets with less supersymmetry (e.g. $\mathcal{N} = (2,1)$, $\mathcal{N} = (2,0)$ and etc.) are discussed a bit in [23, 25, 33], and further properties are yet to be studied.

**Free exotic theories.** Our interests are focused on the exotic fermionic field $\Psi_{MN}$ and it is contained in the covariant field content of all these exotic multiplets. To begin with, we first look at the 6d $\mathcal{N} = (p,q)$ massless little group $G_{\text{little}} = SU(2) \times SU(2) \times G_{(p,q)}^R$, which is the subgroup of $Spin(5,1) \times G_{(p,q)}^R$ preserving a null-momentum vector. Omitting the R-symmetry part $G_{(p,q)}^R$ for a moment, the spacetime little group representation $(\mathbf{4}, \mathbf{1})$ of $SU(2) \times SU(2)$ corresponds to a covariant *chiral* fermionic 2-form-spinor field $\Psi_{MN}$, which we refer to as the aforementioned exotic gravitino. This spinor field is anti-symmetric and its field strength is self-dual and gauge invariant as when introduced in [8]

$$
\Psi_{MN} = -\Psi_{NM} \,,
$$
$$
H_{MNP} \equiv 3\partial_{[M}\Psi_{NP]}, \qquad H = \star H \qquad \text{invariant under} \qquad \delta\Psi_{MN} = 2\partial_{[M}\epsilon_{N]},
\tag{58}
$$

---

[10]See [24] for a discussion on bosonic "higher-biform symmetries" of these exotic gravitons.

where $\epsilon_N$ is an arbitrary vector-spinor. Remarkably, the self-duality equation of $H$ alone is not strong enough to ensure that $\Psi_{MN}$ propagates degrees of freedom in $(\mathbf{4},\mathbf{1})$ of the little group after gauge fixing.[11] One can show [25,39] that the Rarita-Schwinger type field equation (27) for $p = 2$

$$\Gamma^{MNPQR}H_{PQR} = 0\,, \tag{59}$$

together with chirality condition

$$\Gamma_7 \Psi_{MN} = \Psi_{MN}\,, \tag{60}$$

describe the correct little group representation $(\mathbf{4},\mathbf{1})$, and they imply the self-dual condition $H = \star H$. We use $\Gamma_M$ to denote the 6d gamma matrices and $\Gamma_7$ is the chirality matrix. This field equation (59) comes from the free action (24) for $p = 2$, and in particular, the discussions for fermionic $p$-form symmetries apply here. The only difference is that the exotic gravitino is in addition a chiral fermionic field (60). We thus claim that for 6d free exotic theories there is chiral fermionic 2-form symmetry.

**Dimensional reduction of 6d chiral fermionic 2-form to 5d.** The reduction of free chiral fermionic 2-form $\Psi_{MN}$ to five dimensions has been studied in [25] at the level of actions. Following their conventions, $\Gamma_7$ matrix is diagonal in a representation that relates the 6d and 5d gamma matrices block-wisely. The 6d free chiral fermionic 2-form $\Psi_{MN} = \begin{pmatrix} \hat{\psi}_{MN} \\ 0 \end{pmatrix}$ described by the Lagrangian

$$\mathcal{L}_{6\mathrm{d}} = \bar{\Psi}_{MN}\Gamma^{MNPQR}\partial_P \Psi_{QR}\,, \tag{61}$$

yields upon reduction to 5d the following Lagrangian

$$\mathcal{L}_{5\mathrm{d}} = \bar{\psi}_{\mu\nu}\gamma^{\mu\nu\rho\sigma\tau}\partial_\rho\psi_{\sigma\tau} + 2i\bar{\psi}_{\mu\nu}\gamma^{\mu\nu\rho\sigma}\partial_\rho\psi_\sigma - 2i\bar{\psi}_\mu\gamma^{\mu\nu\rho\sigma}\partial_\nu\psi_{\rho\sigma}\,, \tag{62}$$

where the two 5d fields are identified as $\psi_{\mu\nu} = \hat{\psi}_{\mu\nu}$ and $\psi_\mu = \hat{\psi}_{\mu 5}$ ($\mu, \nu = 0,\ldots,4$). There are 5d gauge variations descending from $\delta\hat{\psi}_{MN} = 2\partial_{[M}\epsilon_{N]}$

$$\delta\psi_{\mu\nu} = 2\partial_{[\mu}\epsilon_{\nu]}\,, \qquad \delta\psi_\mu = \partial_\mu\epsilon_5\,, \tag{63}$$

under which the 5d action is invariant. This allows us to define a loop operator

$$V_\eta(\mathcal{C}) = \exp\left(i\oint_{\mathcal{C}}(\bar{\eta}\psi_\mu + \bar{\psi}_\mu\eta)\,dx^\mu\right), \tag{64}$$

and a surface operator

$$W_\xi(\mathcal{S}) = \exp\left(i\oint_{\mathcal{S}}\frac{1}{2}(\bar{\xi}\psi_{\mu\nu} + \bar{\psi}_{\mu\nu}\xi)\,dS^{\mu\nu}\right). \tag{65}$$

Varying the action (62) with respect to $\bar{\psi}_\mu$ and $\bar{\psi}_{\mu\nu}$ give rise to

$$-2i\gamma^{\mu\nu\rho\sigma}\partial_\nu\psi_{\rho\sigma} = 0\,, \qquad \gamma^{\mu\nu\rho\sigma\tau}\partial_\rho(\psi_{\sigma\tau} - 2i\gamma_\sigma\psi_\tau) = 0\,, \tag{66}$$

which lead to a conserved 2-form current

$$\mathcal{J}_{\mu\nu} \equiv 2i\gamma_{\mu\nu\rho\sigma}\psi^{\rho\sigma}\,, \qquad \partial^\nu\mathcal{J}_{\mu\nu} = 0\,. \tag{67}$$

---

[11]This is the opposite of what happened with the 6d bosonic chiral 2-form $B_{MN}$, where the self-dual condition $\star(dB) = dB$ does describe the correct little group representation $(\mathbf{3},\mathbf{1})$ and it implies the usual second order field equation $d \star (dB) = 0$.

and a conserved 3-form current respectively

$$\tilde{\mathcal{J}}_{\mu\nu\rho} \equiv -2\gamma_{\mu\nu\rho\sigma\tau}(\psi^{\sigma\tau} - 2i\gamma^{\sigma}\psi^{\tau}), \qquad \partial^{\rho}\tilde{\mathcal{J}}_{\mu\nu\rho} = 0. \tag{68}$$

We then construct topological operators

$$U_{\epsilon}(\mathcal{V}) = \exp\left(i\int_{\mathcal{V}}\left[\bar{\epsilon}(\star\mathcal{J})_{(3)} + (\star\bar{\mathcal{J}})_{(3)}\epsilon\right]\right), \tag{69}$$

supported on 3d volume $\mathcal{V}$ and

$$\tilde{U}_{\tilde{\epsilon}}(\tilde{\mathcal{S}}) = \exp\left(i\int_{\tilde{\mathcal{S}}}\left[\bar{\tilde{\epsilon}}(\star\tilde{\mathcal{J}})_{(2)} + (\star\bar{\tilde{\mathcal{J}}})_{(2)}\tilde{\epsilon}\right]\right), \tag{70}$$

with surface support $\tilde{\mathcal{S}}$. They act on $V_{\eta}(\mathcal{C})$ and $W_{\xi}(\mathcal{S})$ as

$$\begin{aligned}
\langle U_{\epsilon}(\mathcal{V})V_{\eta}(\mathcal{C})\rangle &= \exp\left(i(\bar{\epsilon}\eta + \bar{\eta}\epsilon)\langle\mathcal{C},\mathcal{V}\rangle\right)\langle V_{\eta}(\mathcal{C})\rangle, \\
\langle \tilde{U}_{\tilde{\epsilon}}(\tilde{\mathcal{S}})W_{\xi}(\mathcal{S})\rangle &= \exp\left(i(\bar{\tilde{\epsilon}}\xi + \bar{\xi}\tilde{\epsilon})\langle\mathcal{S},\tilde{\mathcal{S}}\rangle\right)\langle W_{\xi}(\mathcal{S})\rangle.
\end{aligned} \tag{71}$$

As a conclusion, when compactified on a circle, the six dimensional chiral fermionic 2-form symmetry gives fermionic 2-form symmetry and fermionic 1-form symmetry in five dimensions. Similar result also exists in reduction from ten to nine dimensions: one considers 10d free chiral fermionic 4-form described by the action (24) for $p = 4$, which reduces to a system involving 9d fermionic 4-form and 3-form. For the same reason, 10d fermionic 4-form symmetry leads to 9d fermionic 4-form and 3-form symmetries. However, these fermionic fields do not fit into super Poincaré multiplets.

To make contact with five-dimensional (linearised) supergravity, we can use the second equation of motion (66) to eliminate $\psi_{\mu\nu}$ and obtain a free Rarita-Schwinger Lagrangian for $\psi_{\mu}$ in 5d [25], which exhibits solely fermionic 1-form symmetry. Alternatively, the same equation of motion can be used to integrate out $\psi_{\mu}$ and leave alone a fermionic 2-form action for $\psi_{\mu\nu}$ in five dimensions (it could be an ingredient of dual description of five-dimensional supergravity studied in [40]), which only gives rise to fermionic 2-form symmetry. We also see that this switching between fermionic 1-form and 2-form symmetries has the origin from one dimension higher.

## 4 Gauging of fermionic higher-form symmetries

In this section, we discuss the gauging of fermionic $p$-form global symmetries introduced in this paper. In certain examples (e. g. free fermionic $p$-form field), the gauging of fermionic $p$-form symmetries is obstructed by an 't Hooft anomaly, which can be cancelled by a higher-dimensional anomaly TQFT.

**Free fermionic $p$-form gauge field**  We consider the free fermionic $p$-form gauge field written in the differential form notations (we set the coefficient to be $(-1)$ for simplicity)

$$S[\psi]_{\text{free}} = \int_{M^{(d)}} -\bar{\psi}_{(p)}\wedge\gamma_{(d-2p-1)}\wedge d\psi_{(p)}. \tag{72}$$

To gauge the $p$-form shifting symmetry of $\psi_{(p)}$, we introduce a background $(p+1)$-form fermionic field $\Psi_{(p+1)}$, with the following gauge transformation

$$\begin{aligned}
\delta\psi_{(p)} &= \epsilon_{(p)}, \\
\delta\Psi_{(p+1)} &= d\epsilon_{(p)}.
\end{aligned} \tag{73}$$

After coupling $\Psi_{(p+1)}$ with the $(p+1)$-form current $\mathcal{J}_{(p+1)}$, the action becomes

$$
\begin{aligned}
S[\psi]_{\text{free}} &= \int_{M^{(d)}} -\bar{\psi}_{(p)} \wedge \gamma_{(d-2p-1)} \wedge d\psi_{(p)} + \Psi_{(p+1)} \wedge (\star \mathcal{J}_{(p+1)}) \\
&= \int_{M^{(d)}} -(d\psi_{(p)} - \Psi_{(p+1)}) \wedge (\star \mathcal{J}_{(p+1)}),
\end{aligned}
\tag{74}
$$

which is not gauge invariant under 73 (Note that $\mathcal{J}_{(p+1)}$ is not gauge invariant). Hence the fermionic symmetry has an 't Hooft anomaly.

To better illustrate the point, let us rewrite (72) in terms of an integration of a $(d+1)$-form on $M^{(d+1)}$, where $M^{(d)} = \partial M^{(d+1)}$ is the boundary of $M^{(d+1)}$.

$$
S[\psi]_{\text{free}} = \int_{M^{(d+1)}} -d\bar{\psi}_{(p)} \wedge \Gamma_{(d-2p-1)} \wedge d\psi_{(p)}.
\tag{75}
$$

$\Gamma_{(d-2p-1)}$ is the antisymmetric product of $\gamma$-matrices $\Gamma_i$ in $(d+1)$-dimensions. When $d$ is even, we can use a set of $\Gamma_i$ ($i = 0, \ldots, d$) with the same dimension as the $\gamma$-matrices in $d$-dimensions. When $d$ is odd, $\Gamma_i$ can be chosen as

$$
\Gamma_i = \begin{pmatrix} 0 & \gamma_i \\ \gamma_i & 0 \end{pmatrix} \quad (i = 0, \ldots, d-1), \qquad \Gamma_d = \begin{pmatrix} I & 0 \\ 0 & -I \end{pmatrix},
\tag{76}
$$

and dimensions of the spinors $\Psi_{(p+1)}$ and $\psi_{(p)}$ after the uplift are also doubled.

We use the gauge invariant linear combination $\Psi_{(p+1)} - d\psi_{(p)}$ to write the gauged action on $M^{(d+1)}$ as

$$
S[\psi]_{\text{gauge invariant}} = \int_{M^{(d+1)}} -(\bar{\Psi}_{(p+1)} - d\bar{\psi}_{(p)}) \wedge \Gamma_{(d-2p-1)} \wedge (\Psi_{(p+1)} - d\psi_{(p)}).
\tag{77}
$$

Nonetheless, the extra terms cannot be absorbed into the $d$-dimensional action, hence they are interpreted as an 't Hooft anomaly polynomial. We can also write the extra terms as a $(d+2)$-form (all the spinors are lifted to $(d+2)$-dimensions)

$$
I_{d+2} = -2\bar{\Psi}_{(p+1)} \wedge \Gamma_{(d-2p-1)} \wedge d\Psi_{(p+1)} + d\bar{\Psi}_{(p+1)} \wedge \Gamma_{(d-2p-1)} \wedge d\psi_{(p)} + d\bar{\psi}_{(p)} \wedge \Gamma_{(d-2p-1)} \wedge d\Psi_{(p+1)},
\tag{78}
$$

which is gauge invariant. Unlike the bosonic cases,[12] $I_{d+2}$ contains the matter field $\psi_{(p)}$ as well.

**Fermionic "BF" theory with gaugeable fermionic symmetry**  We discuss the examples of fermionic "BF" theories with a mixed 't Hooft anomaly, where one can gauge a part of fermionic global symmetries.

We construct an action in $d$-dimensions, in the form of

$$
S_{\text{free}} = \int_{M^{(d)}} -\bar{\chi}_{(k)} \wedge \gamma_{(d-p-k-1)} \wedge d\psi_{(p)} + \text{c.c.}
\tag{79}
$$

The theory has a fermionic $p$-form global symmetry

$$
\delta\psi_{(p)} = \lambda_{(p)},
\tag{80}
$$

---

[12]For instance, the mixed anomaly between electric and magnetic 1-form symmetries in the 4d Maxwell theory [1].

and a fermionic $k$-form global symmetry

$$\delta\chi_{(k)} = \rho_{(k)}. \tag{81}$$

They have background gauge fields $\Psi_{(p+1)}$ and $X_{(k+1)}$ respectively.

Similar to the discussion for a free fermionic $p$-form field, one cannot gauge the fermionic $p$-form and $k$-form global symmetries simultaneously, due to a mixed 't Hooft anomaly in $(d+1)$-dimensions. However, one can gauge only one of these symmetries.

For instance, the new action after the gauging the $p$-form symmetry is

$$S_{\text{gauge invariant}} = \int_{M^{(d)}} -\bar{\chi}_{(k)} \wedge \gamma_{(d-p-k-1)} \wedge (d\psi_{(p)} - \Psi_{(p+1)}) + \text{c.c.} \tag{82}$$

It is gauge invariant under

$$\begin{aligned}
\delta\psi_{(p)} &= \epsilon_{(p)}, \\
\delta\Psi_{(p+1)} &= d\epsilon_{(p)}.
\end{aligned} \tag{83}$$

Nonetheless, the gauge symmetry of the field $\chi_k$ is broken after the gauging process.

In the special case of $k=0$, which is a spinor coupled to a fermionic $p$-form field, there is no gauge symmetry to break.

**Other theories with gaugeable fermionic symmetries**  We can also construct actions where all the fermionic fields are accompanied with a derivative, for instance

$$S[\psi]_{\text{free}} = -\int_{M^{(d)}} d\bar{\psi}_{(p)} \wedge *d\psi_{(p)}. \tag{84}$$

In this case, the fermionic $p$-form global symmetry ($d\lambda_{(p)} = 0$)

$$\delta\psi_{(p)} = \lambda_{(p)}, \tag{85}$$

can be gauged by coupling to the background gauge field $\Psi_{(p+1)}$:

$$S[\psi]_{\text{gauge invariant}} = -\int_{M^{(d)}} (\bar{\Psi}_{(p+1)} - d\bar{\psi}_{(p)}) \wedge *(\Psi_{(p+1)} - d\psi_{(p)}). \tag{86}$$

# 5  Examples with broken fermionic symmetries

## 5.1  Rarita-Schwinger field coupled to gauge field

In this section, we discuss an interacting system with a Rarita-Schwinger field $\psi_\mu$ coupled to a gauge field $A_\mu$ and a spinor $\xi$, where the fermionic 1-form symmetry associated to $\psi_\mu$ is broken by the coupling term. The gauge field $A_\mu$ can be Abelian or non-Abelian, and the spinor $\xi$ which plays the role of auxiliary field is also coupled to the gauge field $A_\mu$. This model was firstly introduced in [41] as *extended gauged Rarita-Schwinger theory*, where an exact off-shell fermionic gauge invariance is achieved with help of the auxiliary field $\xi$. This off-shell fermionic gauge invariance is a generalization of the fermionic 0-form gauge symmetry (10), and we replace $\partial_\mu$ by the usual gauge covariant derivative $D_\mu$. In addition, this theory has no superluminal modes and it is consistent as a non-supersymmetric, classical field theory in four dimensions. However, whether it can be properly quantized remains a open question and for more details we refer to [42, 43] and references therein.

We use the following action, which can be defined in general $d$ space-time dimensions, and for $d > 3$:

$$S = -\frac{1}{4} \int d^d x F_{\mu\nu} F^{\mu\nu} - \frac{1}{4} \int d^d x \left( \bar{\psi}_\mu \gamma^{\mu\nu\rho} \overrightarrow{D}_\nu \psi_\rho - \bar{\psi}_\mu \overleftarrow{D}_\nu \gamma^{\mu\nu\rho} \psi_\rho \right)$$
$$+ \frac{ig}{4} \int d^d x \left( \bar{\xi} F_{\mu\nu} \gamma^{\mu\nu\rho} \psi_\rho - \bar{\psi}_\mu \gamma^{\mu\nu\rho} F_{\nu\rho} \xi \right) - \frac{ig}{8} \int d^d x \left( \bar{\xi} F_{\mu\nu} \gamma^{\mu\nu\rho} \overrightarrow{D}_\rho \xi - \bar{\xi} \overleftarrow{D}_\mu \gamma^{\mu\nu\rho} F_{\nu\rho} \xi \right).$$
$$(87)$$

The covariant derivatives are[13]

$$\overrightarrow{D}_\mu \psi_\nu = \partial_\mu \psi_\nu - ig A_\mu \psi_\nu,$$
$$\bar{\psi}_\nu \overleftarrow{D}_\mu = \partial_\mu \bar{\psi}_\nu + ig A_\mu \bar{\psi}_\nu,$$
$$\overrightarrow{D}_\mu \xi = \partial_\mu \xi - ig A_\mu \xi,$$
$$\bar{\xi} \overleftarrow{D}_\mu = \partial_\mu \bar{\xi} + ig A_\mu \bar{\xi}.$$
$$(88)$$

The above action (87) has a fermionic 0-form gauge symmetry

$$\delta_\epsilon A_\mu = 0, \qquad \delta_\epsilon \bar{\psi}_\mu = \bar{\epsilon} \overleftarrow{D}_\mu, \qquad \delta_\epsilon \psi_\mu = \overrightarrow{D}_\mu \epsilon, \qquad \delta_\epsilon \bar{\xi} = \bar{\epsilon}, \qquad \delta_\epsilon \xi = \epsilon, \qquad (89)$$

and the usual 0-form gauge symmetry

$$\delta_\alpha A_\mu = \frac{1}{g} \partial_\mu \alpha, \qquad \delta_\alpha \bar{\psi}_\mu = -i\alpha \bar{\psi}_\mu, \qquad \delta_\alpha \psi_\mu = i\alpha \psi_\mu, \qquad \delta_\alpha \bar{\xi} = -i\alpha \bar{\xi}, \qquad \delta_\alpha \xi = i\alpha \xi. \qquad (90)$$

The equations of motion for $\psi_\mu$, $\xi$ and $A_\mu$ read

$$\frac{1}{2} \gamma^{\mu\nu\rho} \overrightarrow{D}_\nu \left( \psi_\rho - \overrightarrow{D}_\rho \xi \right) = 0,$$
$$-\frac{ig}{4} F_{\mu\nu} \gamma^{\mu\nu\rho} \left( \psi_\rho - \overrightarrow{D}_\rho \xi \right) = 0,$$
$$-\frac{1}{2} \partial_\mu F^{\mu\nu} + \frac{ig}{4} \partial_\mu \left( \bar{\xi} \gamma^{\mu\nu\rho} \psi_\rho - \bar{\psi}_\rho \gamma^{\rho\mu\nu} \xi + \frac{1}{2} \bar{\xi} \overleftarrow{D}_\rho \gamma^{\rho\mu\nu} \xi - \frac{1}{2} \bar{\xi} \gamma^{\mu\nu\rho} \overrightarrow{D}_\rho \xi \right)$$
$$-\frac{ig}{2} \bar{\psi}_\mu \gamma^{\mu\nu\rho} \psi_\rho - \frac{g^2}{4} \bar{\xi} F_{\mu\rho} \gamma^{\mu\nu\rho} \xi = 0.$$
$$(91)$$

Applying $\overrightarrow{D}_\mu$ on the first equation, one gets the second equation by virtue of the antisymmetrization in covariant derivatives, thus the equation of motions of $\xi$ is a subset of those of $\psi_\mu$ and indicating that $\xi$ is auxiliary.

One can construct the following Wilson loop observable that is gauge invariant under both (89) and (90) gauge symmetries:

$$\mathcal{L}_{\eta,\zeta}(\mathcal{C}) = \text{Tr} \exp \mathcal{P} \oint_{\mathcal{C}} \begin{pmatrix} 0 & (\bar{\psi}_\mu - \bar{\xi} \overleftarrow{D}_\mu) \eta \\ \bar{\zeta} (\psi_\mu - \overrightarrow{D}_\mu \xi) & 0 \end{pmatrix} dx^\mu. \qquad (92)$$

We used an off-diagonal $2 \times 2$ matrix form, similar to the fermionic part of Wilson loops in the literature [44, 45]. Nonetheless, the loop operator (92) is explicitly Lorentz covariant.

---

[13]As mentioned, the non-Abelian version of the model is also valid and for simplicity we demonstrate here computations only in the Abelian case.

However, one cannot construct a topological generator for fermionic global 1-form symmetry, as the equations of motion (91) are not in form of a total derivative of $d$. Suppose that we write down the operator

$$U_\epsilon(M^{(d-2)}) = \exp\left( i \int_{M^{(d-2)}} \left[ \bar\epsilon (\star \mathcal{J})_{(d-2)} + (\star \bar{\mathcal{J}})_{(d-2)} \epsilon \right] \right),$$ (93)

$$\mathcal{J}^{\mu\nu} = \gamma^{\mu\nu\rho} \left( \psi_\rho - \vec{D}_\rho \xi \right).$$ (94)

$\mathcal{J}^{\mu\nu}$ does not satisfy the on-shell condition $d \star \mathcal{J} = 0$. $\int_{M^{(d-1)}} d \star \mathcal{J}$ is not a quantized quantity either.

Now we comment on the physical interpretation of the model from the perspective of fermionic 1-form symmetry breaking. When the gauge coupling $g = 0$, the theory (87) is reduced to a free Rarita-Schwinger field with a free Maxwell sector. Hence (93) become topological and the fermionic 1-form symmetry is restored. When one turns on a non-zero $g$, the fermionic 1-form symmetry is explicitly broken. It would be interesting to compute the vacuum expectation value of the loop parameter (92) and analyze its behaviour for different $g$ in the future.

## 5.2 Swampland implications

We briefly discuss how to relate the fermionic global symmetries we discussed in this paper with the no global symmetry swampland conjectures.

Analogous to the bosonic global symmetries, a quantum gravity theory with an exact fermionic global symmetry should live in the swampland. Since one cannot consistently gauge the fermionic shifting symmetry in a fermionic $p$-form field (see section 4), such symmetries should be explicitly broken.

For instance, in the usual formulations of supergravity, the local supersymmetry transformation of $\psi_\mu$ is

$$\begin{aligned}
\delta_\epsilon \psi_\mu &= \nabla_\mu \epsilon \\
&= \partial_\mu \epsilon + \frac{1}{4} w_\mu^{ab} \gamma_{ab} \epsilon,
\end{aligned}$$ (95)

where $w_\mu^{ab}$ is the spin-connection written in matrix form.

If we write down the loop operator for the Rarita-Schwinger field $\psi_\mu$

$$V_\eta(\mathcal{C}) = \exp\left( i \oint_{\mathcal{C}} (\bar\eta \psi_\mu + \bar\psi_\mu \eta) \, dx^\mu \right),$$ (96)

it is not gauge invariant on a general curved background. The fermionic 1-form global symmetries are explicitly broken in supergravity theories, which is consistent with the no global symmetry swampland conjectures.

We can also consider the fate of fermionic 0-form shift symmetry $\delta \psi = \epsilon$ for a spinor field ($d\epsilon = 0$). In a usual supergravity theory, the free action for $\psi$ becomes

$$S[\psi]_{\text{sugra}} = -\int d^d x \sqrt{|\det\{g\}|} \, \bar\psi \gamma^\mu \nabla_\mu \psi,$$ (97)

which breaks the shift symmetry of $\psi$. Of course, one can still ask about the possibilities to take $\epsilon$ to be covariantly constant, i.e. $\nabla_\mu \epsilon = 0$. If the curved space-time admits such spinors then this shift symmetry (with $d\epsilon = 0$ replaced by $\nabla_\mu \epsilon = 0$) is unbroken. Such a global symmetry exists on a space-time with covariantly constant spinors, but it is not preserved

in general curved space-time. Hence we do not have such fermionic global symmetries in a gravity theory.

For a fermionic $p$-form field in curved space-time with action

$$S[\psi] = \int_{M^{(d)}} -\bar{\psi}_{(p)} \wedge \gamma_{(d-2p-1)} \wedge \nabla \psi_{(p)}, \tag{98}$$

one can also introduce the shifting symmetry $\delta \psi_{(p)} = \epsilon_{(p)}$, where the parameter $\epsilon_{(p)}$ satisfies the covariantly flat condition

$$\nabla \epsilon_{(p)} = 0. \tag{99}$$

Nonetheless, such a symmetry is broken on a general space-time manifold as well.

# 6 Discussions

In this paper, we introduced and explored the concept of fermionic $p$-form global symmetries. We discussed physical examples with the symmetry and their gaugings, as well as examples where the fermionic $p$-form global symmetries are broken. Here we clarify some points and discuss the future directions.

- Non-compactness of the fermionic symmetry group

  In the cases of fermionic $p$-form global symmetries, the symmetry parameter is a spinor with Grassmannian components, thus the symmetry group is always a non-compact fermionic translation group. This feature obstructs the partial breaking of such symmetries, since the fermionic charge associated to the symmetry is not quantized. One may need to compactify the space of spinors, in order to construct more non-trivial models with fermionic $p$-form global symmetries.

- SUSY transformation of Wilson loops in SUSY gauge theories

  In SUSY gauge theories, there is a natural fermionic symmetry acting on the Wilson loop objects, which is the SUSY transformation, see e. g. [46]. In our language, such a symmetry is still a fermionic 0-form symmetry that acts on local fields, e. g. $A_\mu$. Since the action of symmetry is not related to the linking of a $(d-2)$-dimensional topological operator with the Wilson loop.

  Similar considerations has been carried out in the context of supergeometry [47], where the authors introduced global $p$-form (super)symmetries generated by tensorial supercurrents. These supercurrents are made out of superforms which are generalisation of ordinary bosonic differential forms, and the additional spinorial components in the superforms are related to the tensor spinor currents that appear in our construction of fermionic higher-form symmetries.

- Non-invertible fermionic higher-form symmetries

  The fermionic topological defect lines (TDL) were recently studied in 2d fermionic CFTs [48], which generate non-invertible 0-form fermionic symmetries. It is tentative to discuss the analogue of fermionic topological defects in 3d or higher dimensional fermionic CFTs, where non-invertible fermionic higher-form symmetries potentially exist.

- String theory realizations

  One may attempt to realize the fermionic $p$-form global symmetries in string theory. Nonetheless, one needs a massless fermionic $p$-form gauge field in the space-time, that

is almost free. Such a scenario only happens in the free, tensionless limit of superstring theory, where we only have an infinite tower of free, massless higher-spin fields. It would be interesting if one can construct a setup that is not a completely free system.

- 6d (4,0) theory

A free theory based on the 6d $\mathcal{N} = (4, 0)$ supermultiplet exists, and its circle reduction yields linearised 5d $\mathcal{N} = 8$ supergravity [8]. The interacting $(4, 0)$ theory is also conjectured to exist, and as an extension of the free $(4, 0)$ tensor theory the interacting one could be the strong coupling limit of of 5d non-linear maximal supergravity (see [38] for a recent review). In this context, the presumed interacting $(4, 0)$ theory would be a new superconformal phase [49] of M-theory at six dimensions with maximal supersymmetry. On the other hand, standard folklore states that there is no global symmetries in quantum gravity. According to this criterion, fermionic 2-form symmetries of the free limit must be either gauged or broken in the UV. Naive gauging can not be applied to free fermionic $p$-form symmetries as shown in section 4. It would be interesting to ask how the gauging/breaking mechanism goes when we moving from the free theory to the interacting version. For instance, given the relation between 5d linearised SUGRA and free $(4, 0)$ theory, one can argue that couplings between $\Psi_{MN}$ and exotic graviton $C_{MNPQ}$ break the fermionic 2-form symmetry explicitly, while the couplings between linearised graviton $h_{\mu\nu}$ and gravitino $\psi_\mu$ break fermionic 1-form symmetry in 5d. Moreover, as already pointed out in [8], extended objects exist and carrying various charges according to the (4,0) superalgebra. We can ask how to break the fermionic 2-form symmetry by these localized UV input.

- Fermionic TQFTs

We have also constructed a novel type of topological quantum field theories (on flat manifolds) using fermionic tensor fields, with an action of the form

$$
S = \begin{cases}
\sum_{p_i+q_i=d-1} c_i \int_{M^{(d)}} \bar{\psi}_{(p_i)} \wedge d\psi_{(q_i)} & (\text{odd } d), \\
\sum_{p_i+q_i=d-1} c_i \int_{M^{(d)}} \bar{\psi}_{(p_i)} \wedge (1 + \gamma_{d+1}) d\psi_{(q_i)} & (\text{even } d).
\end{cases}
\tag{100}
$$

This is beyond the scope of usual spin TQFTs, which only contain 0-form spinors, see for example [50]. In particular, the name "fermionic higher-form symmetry" was also mentioned in [51]. It would be interesting to further investigate the properties of the fermionic tensor TQFTs and their relations with the known models.

# Acknowledgements

We thank Victor Lekeu, Ran Luo, Ruben Minasian, Sakura Schafer-Nameki, Kaiwen Sun, Qing-Rui Wang, Junbao Wu, Fengjun Xu, Zhi-Cheng Yang for discussions.

**Funding information**  YNW and YZ are supported by National Science Foundation of China under Grant No. 12175004 and by Peking University under startup Grant No. 7100603667. YNW is supported by Young Elite Scientists Sponsorship Program by CAST (2022QNRC001). YZ is supported by the Office of China Postdoc Council (OCPC) and Peking University under Grant No. YJ20220018 and by National Science Foundation of China under Grant No. 12305077.

# A  Conventions

We use the "mostly plus" signature for $d$-dimensional Minkowski metric: $\eta = \text{diag}(-, +, \ldots, +)$. Gamma matrices $\gamma_\mu$ ($\mu = 0, \ldots, d-1$) satisfy the anti-commutation relation

$$\{\gamma_\mu, \gamma_\nu\} = 2\eta_{\mu\nu}. \tag{A.1}$$

Hermitian property of gammas is $(\gamma^\mu)^\dagger = \gamma^0 \gamma^\mu \gamma^0$. In $d = 2m$ dimensions the chirality matrix is defined as

$$\gamma_{d+1} = (-i)^{m+1} \gamma_0 \gamma_1 \ldots \gamma_{d-1}. \tag{A.2}$$

For a spinor $\psi$, its Dirac conjugate is $\bar{\psi} = i\psi^\dagger \gamma^0$.

A differential $p$-form $\omega_{(p)}$ is expressed in components as

$$\omega_{(p)} = \frac{1}{p!} \omega_{\mu_1 \ldots \mu_p} dx^{\mu_1} \wedge \ldots \wedge dx^{\mu_p}, \tag{A.3}$$

and its exterior derivative $(d\omega)_{(p+1)}$ is a $(p+1)$-form with components

$$(d\omega)_{\mu_1 \ldots \mu_{p+1}} = (p+1) \partial_{[\mu_1} \omega_{\mu_2 \ldots \mu_{p+1}]}. \tag{A.4}$$

The components of the wedge product of a $p$-form $\omega_{(p)}$ and a $q$-form $\eta_{(q)}$ are

$$(\omega \wedge \eta)_{\mu_1 \ldots \mu_p \nu_1 \ldots \nu_q} = \frac{(p+q)!}{p! q!} \omega_{[\mu_1 \ldots \mu_p} \eta_{\nu_1 \ldots \nu_q]}. \tag{A.5}$$

The Hodge star operator $\star$ maps $p$-forms to $(d-p)$-forms and our convention is

$$(\star \omega)_{\mu_1 \ldots \mu_{d-p}} = \frac{1}{p!} \varepsilon_{\mu_1 \ldots \mu_{d-p}}{}^{\nu_1 \ldots \nu_p} \omega_{\nu_1 \ldots \nu_p}, \tag{A.6}$$

where $\varepsilon_{\mu_1 \ldots \mu_d}$ is the Levi-Civita symbol and $\varepsilon_{01 \ldots d-1} = 1$, $\varepsilon^{01 \ldots d-1} = -1$.

On a curved manifold with metric $g_{\mu\nu}$, the Levi-Civita symbol $\varepsilon$ generalises to a tensor according to the normalisation

$$\varepsilon_{01 \ldots d-1} = \sqrt{|\det\{g\}|}, \qquad \varepsilon^{01 \ldots d-1} = \frac{-1}{\sqrt{|\det\{g\}|}}. \tag{A.7}$$

The invariant volume form is

$$\sqrt{|\det\{g\}|} \, d^d x \equiv \sqrt{|\det\{g\}|} \, dx^0 \wedge \ldots \wedge dx^{d-1} = \frac{1}{d!} \varepsilon_{\mu_1 \ldots \mu_d} dx^{\mu_1} \wedge \ldots \wedge dx^{\mu_d}, \tag{A.8}$$

here we would also write $dV^{\mu_1 \ldots \mu_d} \equiv dx^{\mu_1} \wedge \ldots \wedge dx^{\mu_d}$ for short.

Integration of $d$-forms over the manifold $M^{(d)}$ is given as

$$\int_{M^{(d)}} v_{(d)} = \int_{M^{(d)}} \frac{1}{d!} v_{\mu_1 \ldots \mu_d} dx^{\mu_1} \wedge \ldots \wedge dx^{\mu_d} = \int_{M^{(d)}} \frac{1}{d!} v_{\mu_1 \ldots \mu_d} dV^{\mu_1 \ldots \mu_d}$$

$$= \int_{M^{(d)}} v_{01 \ldots d-1} d^d x \tag{A.9}$$

$$\equiv \int_{M^{(d)}} v(x)_{01 \ldots d-1} dx^0 dx^1 \ldots dx^{d-1},$$

and integration of a scalar (0-form) $\phi$ is defined as the integral of its Hodge dual

$$\int_{M^{(d)}} \star \phi = \int_{M^{(d)}} \phi \sqrt{|\det\{g\}|} \, d^d x. \tag{A.10}$$

Useful formulae:

$$\star\omega\wedge\eta = \star\eta\wedge\omega = \frac{1}{p!}\omega_{\mu_1\ldots\mu_p}\eta^{\mu_1\ldots\mu_p}\sqrt{|\det\{g\}|}\,d^d x\,,$$
$$d\star v\wedge\omega = (-1)^{d-p-1}\frac{1}{p!}\partial_\mu v^{\mu\nu_1\ldots\nu_p}\omega_{\nu_1\ldots\nu_p}\sqrt{|\det\{g\}|}\,d^d x\,, \tag{A.11}$$

where $\omega$ and $\eta$ are both $p$-forms and $v$ is a $(p+1)$-form.

Let $\mathcal{C}^{(p)}$ denote a $p$-cycle (closed oriented submanifold of dimension $p$), its Poincaré dual is the cohomology class of a $(d-p)$-form $J_{(d-p)}(\mathcal{C}^{(p)})$ such that for any $p$-form $A_{(p)}$ the following relation holds

$$\int_{\mathcal{C}^{(p)}} A_{(p)} = \int_{M^{(d)}} J_{(d-p)}(\mathcal{C}^{(p)})\wedge A_{(p)}\,. \tag{A.12}$$

For submanifolds $U$ and $V$ with dimension $p$ and $d-p-1$ and such that $V$ is the boundary of a $(d-p)$-dimensional submanifold $W$, i.e. $\partial W = V$ (in fact, both $U$ and $V$ should be boundaries of some other submanifolds in order to define the linking number [14]). The linking number $\langle U, V\rangle$ is given as the intersection number $\mathcal{I}(U,W)$ of $U$ and $W$

$$\langle U, V\rangle = \mathcal{I}(U,W) = \int_{M^{(d)}} J_{(d-p)}(U)\wedge J_{(p)}(W) = \int_U J_{(p)}(W)\,. \tag{A.13}$$

This agrees with the definition of [52].

The free fermionic $p$-form action (24) is

$$S[\psi_{(p)}] = -(-1)^{\frac{p(p-1)}{2}}\int d^d x\,\bar\psi_{\mu_1\mu_2\ldots\mu_p}\gamma^{\mu_1\mu_2\ldots\mu_p\,\nu\rho_1\rho_2\ldots\rho_p}\,\partial_\nu\psi_{\rho_1\rho_2\ldots\rho_p}$$

$$= -(-1)^{\frac{p(p-1)}{2}}(-1)^p\int d^d x\,\bar\psi_{\mu_1\mu_2\ldots\mu_p}\partial_\nu\mathcal{J}^{\nu\mu_1\mu_2\ldots\mu_p}$$

$$= -p!(-1)^{\frac{p(p-1)}{2}}(-1)^p(-1)^{p(d-p)}(-1)^{d-p-1}\int\bar\psi\wedge(d\star\mathcal{J})$$

$$= -C(d,p)\int\bar\psi\wedge(d\star\mathcal{J})\,,$$

with

$$C(d,p) = -p!(-1)^{\frac{(p+1)(2d-p)}{2}}\,. \tag{A.14}$$

The shifted $p$-form action used in section 2 $\left(\bar{\mathcal{J}}_{\mu_1\ldots\mu_p\nu} = -\bar\psi^{\rho_1\ldots\rho_p}\gamma_{\rho_p\ldots\rho_1\nu\mu_p\ldots\mu_1}\right)$

$$S[\psi_{(p)}-\epsilon J_{(p)}]$$

$$= S[\psi_{(p)}]+(-1)^{\frac{p(p-1)}{2}}\int d^d x\left((-1)^p\bar\epsilon J_{\mu_1\ldots\mu_p}\partial_\nu\mathcal{J}^{\nu\mu_1\mu_2\ldots\mu_p}+(-1)^p\partial_\nu\bar{\mathcal{J}}^{\nu\rho_1\ldots\rho_p}\epsilon J_{\rho_1\ldots\rho_p}\right)$$

$$= S[\psi_{(p)}]+(-1)^{\frac{p(p-1)}{2}}\int p!(-1)^p(-1)^{p(d-p)}(-1)^{d-p-1}J\wedge\left(d\star[\bar{\mathcal{J}}\epsilon+\bar\epsilon\mathcal{J}]\right)\,,$$

$$= S[\psi_{(p)}]+C(d,p)\int J\wedge\left(d\star[\bar{\mathcal{J}}\epsilon+\bar\epsilon\mathcal{J}]\right)\,,$$

suggesting that $U_\epsilon(M^{(d-p-1)})$ should be

$$U_\epsilon(M^{(d-p-1)}) = \exp\left(i\,C(d,p)\int_{M^{(d-p-1)}}\left(\star[\bar{\mathcal{J}}\epsilon+\bar\epsilon\mathcal{J}]\right)\right)\,.$$

# B  VEV of the fermionic Wilson loop

In this section we give a brief discussion of the vacuum expectation value (VEV) $\langle V_\eta(\mathcal{C})\rangle$ of the fermionic Wilson loop (11). As mentioned, the free action (7) for $\psi_\mu$ in $d = 3$ dimensions solely captures topological degrees of freedom. This is due to the fact that the equations of motion lead to the vanishing of field strength. Consequently, in this particular scenario, VEV can be conveniently normalised as $\langle V_\eta(\mathcal{C})\rangle = 1$, disregarding divergent terms that signify self-interactions. This omission of divergent terms is in line with the common practice in Abelian Chern-Simons theory. When one moves away from the *critical* dimension (i. e. $d = 2p + 1$), the behaviour of $\langle V_\eta(\mathcal{C})\rangle$ becomes similar to the Wilson loop in Maxwell theory.

For instance, in $d = 4$, we introduce the source $J^\mu(x) \equiv \eta \oint_\mathcal{C} dy^\mu \delta(x - y)$ to rewrite $\langle V_\eta(\mathcal{C})\rangle$ as the $J^\mu$-sourced path integral

$$
\begin{aligned}
\langle V_\eta(\mathcal{C})\rangle &= \int \mathcal{D}\psi_\mu \mathcal{D}\bar{\psi}_\mu e^{iS[\psi_\mu,\bar{\psi}_\mu]+i\int_\mathcal{C}(\bar{\eta}\psi_{(1)}+\bar{\psi}_{(1)}\eta)} \\
&= \int \mathcal{D}\psi_\mu \mathcal{D}\bar{\psi}_\mu e^{iS[\psi_\mu,\bar{\psi}_\mu]+i\int_{M^{(4)}} d^4x \,(\bar{J}_\mu\psi^\mu+\bar{\psi}_\mu J^\mu)} \\
&= Z[J_\mu,\bar{J}_\mu].
\end{aligned}
\tag{B.1}
$$

To evaluate the partition function $Z[J_\mu,\bar{J}_\mu]$ in the presence of source $J_\mu$, we need to use the free Rarita-Schwinger propagator, which in momentum space takes the *reverse index* form in suitable gauge [53]

$$
S_{\mu\nu}(p) = -\frac{i}{2}\frac{\gamma_\nu \slashed{p}\gamma_\mu}{p^2}.
\tag{B.2}
$$

Insert the propagator in (B.1) and use the explicit delta-function expression of the source, we have the following result[14]

$$
\langle V_\eta(\mathcal{C})\rangle = Z[J_\mu,\bar{J}_\mu] = \exp\left[-\bar{\eta}\eta \oint_\mathcal{C} dx^\mu \oint_\mathcal{C} dy^\nu \frac{\gamma_\nu\gamma_\rho\gamma_\mu(x-y)^\rho}{4\pi^2(x-y)^3}\right].
\tag{B.3}
$$

The gamma matrices are numerical constants and thus the VEV $\langle V_\eta(\mathcal{C})\rangle$ exhibits a perimeter law.

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
