# Peer review of "Fermionic Higher-form Symmetries"

_SciPost Physics, doi:SciPost Phys. 15, 142 (2023)_

## Round 1 · Referee Report · Anonymous (Referee 1) · 2023-5-25

Strengths
Report
Requested changes
I request one important change, plus some minor corrections/clarifications.
The important aspect to address is the discussion on 't Hooft anomalies. In footnote 1 and throughout the paper, the authors work in flat Minkowski spacetime. In section 4, they encounter an obstruction to gauging, and call it a 't Hooft anomaly. However, anomalies typically describe the non-conservation of a current due to some characteristic class (topological invariant) appearing on the right-hand side of $d \ast j \ne 0$. One should not be able to detect them in flat space, where the bundles trivialize. The authors should clarify the language and how the obstruction they observe is related to the standard notion of anomaly.
A possibly related comment: it is observed in section 4 that the fermionic higher-form symmetry can be gauged only in theories whose Lagrangian involves $d \psi$ but not $\psi$. They provide the examples of $d \psi \wedge \ast d \psi$ (can be gauged) versus $d \psi \wedge \psi \wedge \gamma$ (cannot). This is very reminiscent of what happens with bosonic gauge fields: in Maxwell theory, the Lagrangian depends on $F=dA$ but not on $A$, and the 1-form symmetry is gauged shifting $F$ appropriately. This would not work in e.g. Chern-Simons theory, because the term $dA \wedge A$ cannot be made invariant by a shift of $F$. It would be enlightening if the authors can explain or comment on this analogy.
A list of minor points: 1. page 3 "in section 2 we introduce the notions" --> "notion" 2. page 4. By the definition of $G$ above (2.1), I understand that the fermionic symmetry cannot be a finite group. If this is the case, it would be better to state this fact explicitly, since for now this is only mentioned in the outlook. 3. In (2.11), $\eta$ should be a fermionic analog of the charge of a Wilson loop. It should be defined explicitly 4. In footnote 5, isn't the definition of the authors just the usual one? 5. Below (2.23), the sentence "there is no electromagnetic type dualities between p-form fermionic". Are the authors claiming a no-go theorem, or do they mean that there is no reason to expect such a duality? If the latter, it would be convenient to nuance a bit the statement. 6. In (3.16) and (3.17), state that $\mathcal{C}$ is a line and $\mathcal{S}$ is a surface. 7. In (4.1) and subsequent, the notation $M^{(d)}$ of earlier sections is changed to$M_d$. 8. In (4.5) the authors write the action as $S_{\rm gauged}$, but then they state that there is an anomaly that prevents the gauging. I suggest to replace the notation into a less misleading one, e.g. $S_{\rm gauge~invariant}$

---

## Round 1 · Referee Report · Anonymous (Referee 2) · 2023-6-11

Strengths
Weaknesses
Report
This is a generalization of supersymmetry where the supercharges are supported on codimension >1 loci in spacetime.
It is a possibility that I know has been considered by many people, but I have not previously seen any discussion of it in the literature (in more than two spacetime dimensions).
The authors exhibit several gaussian theories that exhibit such symmetries.
They make the point that such symmetries of these free theories seem to be explicitly broken in coupling to a general curved metric, and therefore do not provide counterexamples to various speculations about quantum gravity. A small comment about the logic: It is true that the usual way of coupling to curved space breaks these symmetries, but the authors didn't seem to show that there is no possible way of writing a covariant action that respects these symmetries.
An interesting and nontrivial class of examples with such symmetry discussed here is fermionic versions of BF theory.
The authors should say a few more words about the canonical quantization and spectrum of these theories (or give a reference) -- as the authors claim in passing, they are topological, in sharp contradistinction to the gapless example of the free Rarita-Schwinger field.
It would be useful to comment on a possible lattice realization of this theory, in the same way the ordinary BF theory (with appropriate coefficient) is a low energy description of the toric code ($Z_n$ gauge theory).
Specifically, it is not clear to me what is the relation between this theory and various fermionic generalizations of the toric code, such as this one:
https://arxiv.org/abs/1309.7032
-- Before equation 2.3:
it says "As another simple example,"
but this is the first example presented.
Requested changes
see above.

---

## Round 1 · Referee Report · Anonymous (Referee 3) · 2023-6-22

Strengths
Weaknesses
Report
Requested changes
Questions
1) Does the fermionic analog of the Wilson loop defined in (2.11) carry physical meaning? It is a meaningful order parameter for the the fermionic symmetry? If this is the case, could you please comment about spontaneous symmetry breaking scenario.
2) Concerning the BF-like theories discussed in Sec. 3.1, I have some questions:
2a) These theories are gapped?
2b) What are the topological indicators? In the usual BF theories in 2+1 for example, the ground state degeneracy depends on the topology of the manifold.
2c) Is there a fermionic analog of the bulk-edge correspondence? In this case, what would be the edge theory?
3) What kind of constraints follow from the existence of 't Hooft anomalies discussed in Sec. 4?

---

## Round 2 · Referee Report · Anonymous (Referee 1) · 2023-7-4

Report

The authors have exhaustively addressed all the minor points raised in the previous report. The major point about ´t Hooft anomalies has been clarified in one example, but not in full generality.
The main problem that remains is that the general construction and analysis assumes to work in flat space, while the analysis of ´t Hooft anomalies requires placing the theories on a non-trivial topology $M^{(d)}$. While the expanded discussion seems to justify the procedure for a free $p$-form spinor, it is still unclear why this step is justified in general.
The issue is the following: the authors observe an "obstruction to gauging", and (understandably) refer to it as a ´t Hooft anomaly. However, while there exists a mathematical definition and classification of anomalies based on cohomology, it does not seem to apply to the present case in general, because the entire construction of the authors is in flat space and cohomology is trivial.
I acknowledge that a comprehensive characterization of this "fermionic" version of ´t Hooft anomalies is beyond the scope of the work, but the authors should more carefully distinguish throughout Section 4 between what is derived and what is a proposal justified heuristically by analogy with known anomalies.

Requested changes

Explicitly clarify the usage of the phrasing "´t Hooft anomaly" and the tension with the fact of working in flat space in Section 4.

  • validity: -
  • significance: -
  • originality: -
  • clarity: -
  • formatting: -
  • grammar: -

Author:  Yinan Wang  on 2023-07-28  [id 3849]

(in reply to Report 1 on 2023-07-04)
Category:
answer to question

Dear referee,

We would like to clarify the notion of 't Hooft anomaly we used in the paper. In general, there are two types of anomalies: (1) local anomalies, which arise from infinitesimal gauge transformations, and is mathematically described by anomaly polynomials and descent formalism; (2) global anomalies, which come from large gauge transformations, and is mathematically described by group cohomology.
The 't Hooft anomalies discussed in the paper belong to local anomalies, hence it does not need to be assigned with a non-trivial group cohomology class.

Sincerely,
The authors

---

## Round 2 · Referee Report · Anonymous (Referee 3) · 2023-7-22

Report

I would like to thank the authors for their efforts in answering all the questions. As far as I can understand, the authors have partially addressed the questions I raised in the previous report, but some important points remain unclear. In particular, the behavior of the vacuum expectation value of the fermionic Wilson is not analyzed. As the authors themselves state in the manuscript, this is left for future work. In principle, this analysis should be possible at least in the case of free fermionic theories. In addition, the added discussion about 't Hooft anomalies around Eq. (4.7) does not address the question about the type of constraints that follow from the existence of such anomalies in the IR sector. In this way, it is seems to be hard to extract much physical information from fermionic higher-form symmetries. I think it would be enlightening to discuss more about these points.
  • validity: -
  • significance: -
  • originality: -
  • clarity: -
  • formatting: -
  • grammar: -

Author:  Yinan Wang  on 2023-07-28  [id 3848]

(in reply to Report 2 on 2023-07-22)
Category:
remark
answer to question

Dear referee,

Thanks for the comment on the vev of fermionic Wilson loops. We will add an appendix on the computation of fermionic Wilson loop vevs for the examples of free Rarita-Schwinger fields in the next version.
For the 't Hooft anomaly polynomial, it is quite unconventional as it involves both the gauge field and matter field. We do not have more detailed comments on the UV/IR matching at this stage, and we would like to clarify it in the future.

Sincerely,
The authors

---

## Round 2 · Author Response

Dear Editor,

We have made significant changes to the draft, addressing many points by the referees. In particular, we expanded on the following physics points:

(1) The gauging of fermionic symmetries and 't Hooft anomalies;

(2) The presence of magnetic fermionic symmetries;

(3) The physical details of the new fermionic TQFTs proposed in the paper, including the gapped properties and edge modes.

We have also added a number of physical explanations regarding the questions of the referees. For the other follow-up questions, such as a full exploration of such symmetries on curved space-time or a lattice, we will investigate them in future works.

Sincerely,
The authors

---

## Round 2 · List of Changes

We have made the following changes to the manuscript according to each referee's request.

Referee 1:

(1) We have expanded Section 4 on the gauging of fermionic symmetries for a free fermionic p-form gauge field. We showed explicitly around (4.3) that the d-dimensional action is not gauge invariant, and there's a 't Hooft anomaly. We also added the I_{d+2} 't Hooft anomaly polynomial in (4.7), which is different from the bosonic cases. Note that it cannot be written in terms of a characteristic class of the background gauge field itself.

(2) On the minor point 2, we added a sentence in the first paragraph of page 4, explaining that the fermionic symmetry group is taken as R^s, where s is the number of spinor components.

(3) On the minor point 3, we added explanations below (2.11) about the parameter \eta, which is an unquantized "charge" of the fermionic Wilson loop.

(4) On the minor point 5, we changed the statement to that there exists a topological generator for the magnetic symmetry, but the existence of EM duality in fermionic theories is unknown, and we do not find a charged object.

(5) We also revised the draft according to the other minor points.

Referee 2:

(1) On page 12 above (3.24), we comment on the spectrum of such fermionic TQFTs. They are gapped analogous to the bosonic counter parts, because the e. o. m. gives rise to pure gauge.

Referee 3:

(1) We commented on the interpretation of fermionic Wilson loop as order parameters at the end of Section 5.1. Nonetheless, the full quantization of such a physical model is still unknown.

(2) We commented on page 12 above (3.24) that such TQFTs are gapped.

(3) On page 12 below (3.22), we discussed the edge modes of the fermionic TQFT in a 3d example.

(4) We expanded the discussion of gauging and 't Hooft anomaly on page 16.

We have also added some references.

---

## Round 3 · Author Response

Dear Editor,
According to Referee 2's comment, we have added a short Appendix B on the vacuum expectation value of the fermionic Wilson loops in the free field case. We have also responded to the referees' questions by comments on the SciPost webpage.
Best regards,
The authors
According to Referee 2's comment, we have added a short Appendix B on the vacuum expectation value of the fermionic Wilson loops in the free field case. We have also responded to the referees' questions by comments on the SciPost webpage.
Best regards,
The authors

---

## Round 3 · List of Changes

We added a short Appendix B on the vacuum expectation value (VEV) of the fermionic Wilson loops. In particular, we gave an example of a free Rarita-Schwinger field in 4d, where we computed the VEV of the fermionic Wilson loop and confirmed that it obeys the perimeter law.

---

## Editorial Decision

published